

# Analysis of a 30 GW offshore wind power scenario in Norway using time-series computed from numerical weather model data

Harald G Svendsen  and John Olav Giæver Tande

SINTEF Energy Research, Trondheim, Norway

**Correspondence:** Harald G Svendsen  (harald.svendsen@sintef.no)

**Abstract.** This article investigates implications of integrating 30 GW offshore wind in Norway. Wind power time series for relevant locations are analysed using 30 years of hourly numerical weather model reanalysis data. The study presents key statistical properties of the wind power time series. The emphasis lies on correlation, geographical smoothing, and variability across different time scales. These findings hold significant relevance for the strategic planning of offshore wind farm development, and for effectively preparing the energy system to accommodate this extensive wind power deployment that would mean a doubling of the Norwegian electricity generation.

## 1 Introduction

The Norwegian government has announced plans to open areas for 30 GW of offshore wind power capacity by 2040. This amounts to an annual power production of about 140 TWh, which is larger than today's annual power consumption of about 135 TWh. Successful integration of all this variable wind power with the Norwegian power system is therefore a matter of great importance that can only be achieved with a good understanding of the characteristics of the offshore wind power.

The main challenge with integrating large amounts of wind power is its variability, a challenge that is exasperated by co-variability of wind power located close to each other. Two key integration issues are high power ramp rates (MW/hour) and extended periods with low wind. These give demanding requirements for the activation speed and the size of needed power balancing resources such as power exchange to neighbouring countries, backup generation capacity, energy storage and/or flexible demand.

The offshore wind power plans in Norway is part of a bigger picture with large-scale wind power development in Northern Europe, so relying on power exchange with neighbouring countries to balance out the variations is unlikely to be sufficient.

In this article we describe analyses based on power time series obtained from publicly available numerical weather model reanalysis data. To some extent, it is an update of previous analysis by Korpås et al. (2007); Tande and Vogstad (1999). Reanalysis data is known to capture overall variability well, but not so good on the exact timing of wind speed fluctuations (Mehrens et al., 2016).

A recent study by Solbrekke et al. (2020) uses a set of observed hourly wind speed data from five Norwegian offshore locations over a period of 16 years to quantify the potential of collective offshore wind power production. A report by Koestler et al. (2020) at the Norwegian Water and Energy directorate (NVE) has looked at how renewable power production is a



challenge for the power systems, based on ERA5 data from 1979-2019. That study is based on existing *onshore* wind farm locations only. Challenges with the correlation of earlier proposed Norwegian wind sites was addressed by Hjelmeland and Nøland (2023), together with a proposal for wind capacity distribution to minimise overall variance. Another recent study considers wind power output and correlation matrix for the 29 biggest offshore wind farms in Europe, estimated based on ERA5 reanalysis data (Grothe et al., 2022). Using the concept of covariance the fact that wind power tends to drive power prices down, Vrana and Svendsen (2024) introduced a way to illustrate the value of additional offshore wind power capacity at different locations in future scenarios.

There are several reanalysis data sets that are publicly available. A recent study by Murcia et al. (2022) assesses the accuracy of different data sets for wind generation simulations in large scale scenarios. Several wind power time series are also available from previous studies, e.g., Gonzales Aparico et al. (2016); Solbrekke and Sorteberg (2022); Koivisto and Leon (2022).

In the following sections we describe which data and main assumptions are used (Section 2), the geographical smoothing of combined wind power output (Section 3, wind power correlation for different locations (Section 4), different characteristics of the temporal variability (Section 5), with some brief concluding remarks (Section 6).

The main novelty of this study is the scenario that is being considered: 30 GW offshore wind in Norway in line with government plans, and located according to recently published areas for potential offshore wind development in Norway. Although the time series analysis methods are well-known, we believe the results to be of wider interest, with supporting material (Svendsen, 2023) that includes Python code used for downloading data and creating all the figures presented in this article. This makes the results easily reproducible and adaptable for other cases.

## 2 Data and assumptions

### 2.1 Wind power sites

The Norwegian Energy and Water Directorate (NVE) has recently identified 20 potential areas for offshore wind development in Norway (NVE, 2023), as an update to the 15 areas announced in 2012 (Berg et al., 2012).

In this study, we consider wind power at all the new areas, and the resulting combined power output if 30 GW is distributed equally in these areas, with 1.5 GW at each site. For part of the analysis we have included a British, a Danish, and a German offshore wind farm site as well. See Figure 1 for a map of all these locations, and Table 1 for names and coordinates used.

Two areas are currently open for development in Norway: *Sørlige Nordsjø II* which belongs to area 18 (Sørvest F) and Utsira Nord which belongs to area 12 (Vestavind F).

The total surface area of the identified sites is $55\,371\,\mathrm{km}^2$. If fully developed with installed power density of $7.5\,\mathrm{MW/km}^2$ this would give total capacity $415\,\mathrm{GW}$. So clearly not all areas will be used for the 30 GW target. Correspondingly, 30 GW would require an area of $4000\,\mathrm{km}^2$, which is $7\,\%$ of the total area identified. Which area will be selected and in which order they will be opened is impossible to say at present.



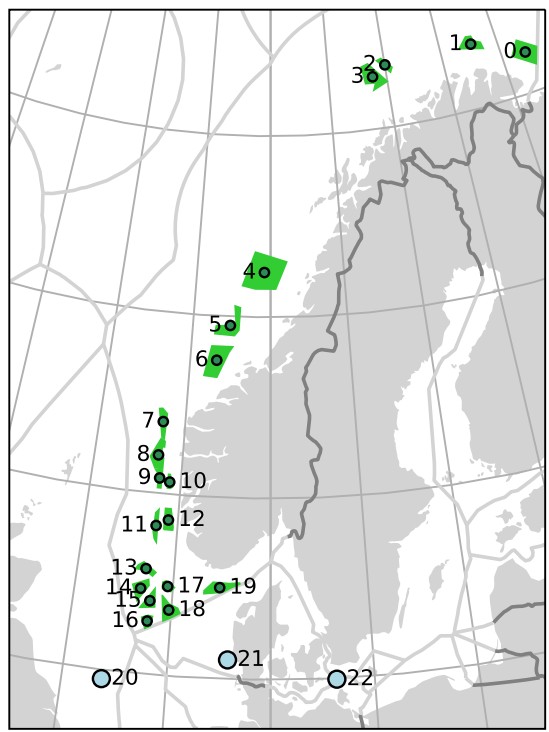

**Figure 1.** The 20 Norwegian offshore wind farm sites (0-19), and 3 other wind farms (20-21) ordered from north to south.

## 2.2 Wind power time series

As mentioned previously, multiple wind resource data sets are available for the creation of power time series. For Norway, the higher-resolution NORA3-WP (Solbrekke and Sorteberg, 2022) would likely be a good choice, but we chose to work with wind speed data from MERRA 2 (Molod et al., 2015) because of its ease of use via the *Renewables ninja* website (Pfenninger and Staffell, 2024).

Wind speeds covering the 30 year period from 1991 to 2020 have been used. Wind speeds at 100 m height have been converted to wind power using an effective wind farm power curve obtained by applying a Gaussian filter with standard deviation $\sigma = 0.2$ on a single wind turbine power curve (Staffell and Green, 2014; Staffell and Pfenninger, 2016). see Figure 2. This is done directly by Renewables ninja. The turbine assumed in our case is the Vestas V80 2000, although the precise choice of wind turbine power curve does not significantly affect the results presented here.

The Gaussian filtering method to obtain wind farm power curves is a crude simplification and the resulting wind power time series are an approximation. It will not give a reliable prediction of annual energy production. However, the variability from hour to hour and correlations between different locations, which are our main interests here, are still considered to be well captured by this approach.





**Table 1.** Wind farm sites

| # | Name | Lat | Lon |
|---|------|-----|-----|
| 0 | Nordavind A | 71.11 | 32.01 |
| 1 | Nordavind B | 71.78 | 27.75 |
| 2 | Nordavind C | 71.73 | 20.01 |
| 3 | Nordavind D | 71.45 | 18.8 |
| 4 | Nordvest A | 66.23 | 9.58 |
| 5 | Nordvest B | 64.75 | 7.41 |
| 6 | Nordvest C | 63.78 | 6.65 |
| 7 | Vestavind A | 61.98 | 3.71 |
| 8 | Vestavind B | 61.06 | 3.62 |
| 9 | Vestavind C | 60.43 | 3.81 |
| 10 | Vestavind D | 60.34 | 4.38 |
| 11 | Vestavind E | 59.11 | 3.85 |
| 12 | Vestavind F (UN) | 59.29 | 4.49 |
| 13 | Sørvest A | 57.9 | 3.54 |
| 14 | Sørvest B | 57.35 | 3.36 |
| 15 | Sørvest C | 57.03 | 3.89 |
| 16 | Sørvest D | 56.46 | 3.84 |
| 17 | Sørvest E | 57.46 | 4.73 |
| 18 | Sørvest F (SN2) | 56.82 | 4.87 |
| 19 | Sønnavind A | 57.52 | 7.39 |
| 20 | GB_Doggerbank | 54.75 | 1.92 |
| 21 | DK_Horns Rev | 55.53 | 7.91 |
| 22 | DE_Baltic2 | 54.97 | 13.18 |

## 2.3 Comparison with actual wind power for coastal site

Before proceeding to the main analysis part, we make another comparison to shed light on the modelling and uncertainties.

The Norwegian Water and Energy Directorate (NVE) publishes wind power production data with an hourly resolution for wind farms in Norway (NVE, 2022). There are no offshore wind farms, and the closest to offshore conditions is probably the

150 MW Smøla wind farm which is located at the coast south west of Trondheim (near site 6 (Nordvest C) in the map) on a relatively flat island. So we select this one and compare the estimated wind power time series for this location (assuming 70 m hub height and Siemens SWT 2.3 turbines) to its reported actual wind power production in the period 2016–2021.

Figure 3 shows a few days as an example from the time-series comparing actual and modelled power output. The overall variability is well captured by the model, whilst the details are not. This is not surprising with the lack of detail in the wind

farm modelling. Notably, wake effects are not considered in the wind power estimation, such that the modelled wind power

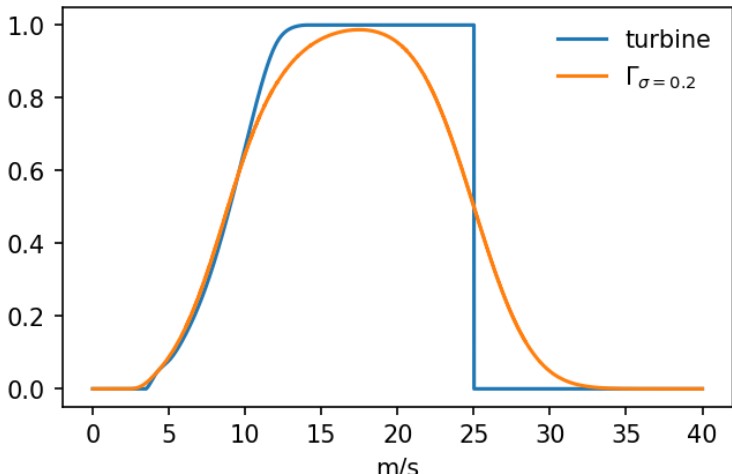

**Figure 2.** Wind farm power curve.

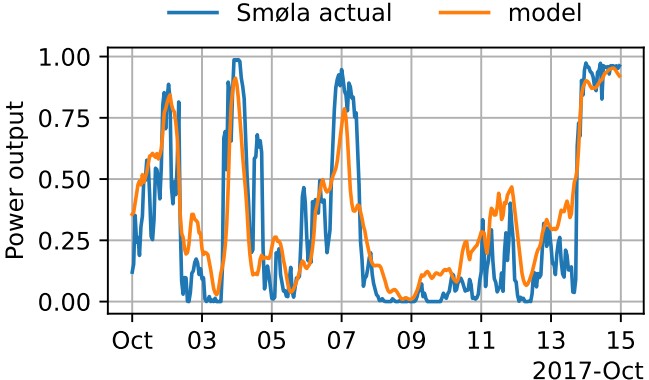

**Figure 3.** Wind power output at Smøla. Comparison between model and actual data

will over-estimate power output. In terms of capacity factors (i.e. mean power output divided by capacity) we find a value of 0.31 in the model vs. 0.25 in the actual data.

Some of the differences are clear from Figure 4 which shows duration curves for wind power output as given by the actual measurement data compared to what the model gives. The biggest difference is that the actual wind farm has much higher 85 occurrence of low wind power output. For example, the model indicates that wind power exceeds 20% of capacity 55% of the time, whereas the actual data shows this to be only 38% of the time.

It is difficult to say exactly what the reason for these differences are. One issue that has not been accounted for is the possibility that not all wind power capacity has been available all the time, e.g. due to faults or maintenance work. Such events would reduce the observed power output from the wind farm.



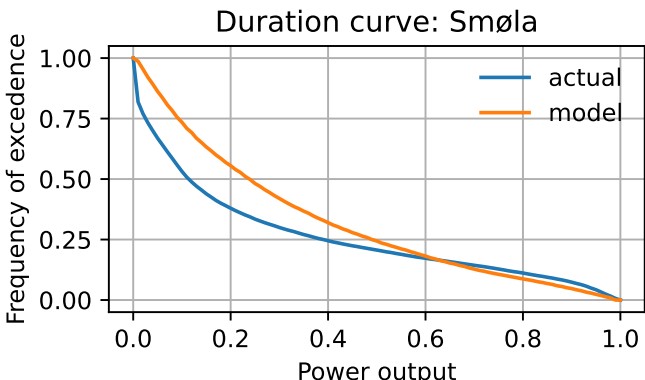

**Figure 4.** Comparison of duration curves for actual wind power vs modelled wind power output

These are significant differences that should act as a reminder that site-specific and wind farm specific issues are important, and that using weather model reanalysis data with a simple power curve to estimate wind power production is indeed a crude approximation. However, the model analysis shows comparable characteristics and provide important information.

## 3 Combined power output

Returning to the 30 GW offshore wind scenario, Figure 5 shows a histogram of the power output from a single wind farm
and the combined output from all 20 wind farms, considering uniform distribution of installed capacity as described earlier. The histograms are markedly different: For an individual wind farm, high and low wind power output situations are common, whereas for the combined output, mid-range power output is more frequent and maximum or minimum power output is rare.

The duration curve in Figure 6 shows the same information in a different form, now including all individual wind farms as separate curves. In this plot, the curves indicate how often the power output exceeds a given value.
As expected, there is a clear smoothing effect for the combined power output, with values more often in the mid-range between about 30% and 80%. From the combined curve, we see that the power output exceeds 30% for about 90% of the time.

This smoothing is a good thing for the power system. Still, even for the combined power output, there are times with very low ($< 0.1$) values. In Section 5.2 we explore such low wind periods in more detail.

## 4 Correlations

In this section, we investigate correlation coefficients of wind power from the various sites.

The correlation coefficient $c_{ij}$ between two time-series $x_i(t)$ and $x_j(t)$ is defined as

$$c_{ij} = \frac{\langle (x_i(t) - \bar{x}_i)(x_j(t) - \langle x_j \rangle) \rangle}{\sigma_i \sigma_j},$$ (1)

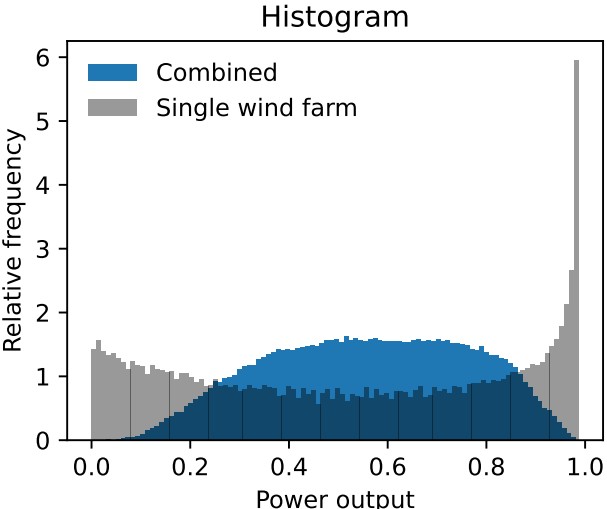

**Figure 5.** Histogram of wind power output from an individual wind farm (#6) and the combined output from all 20 sites.

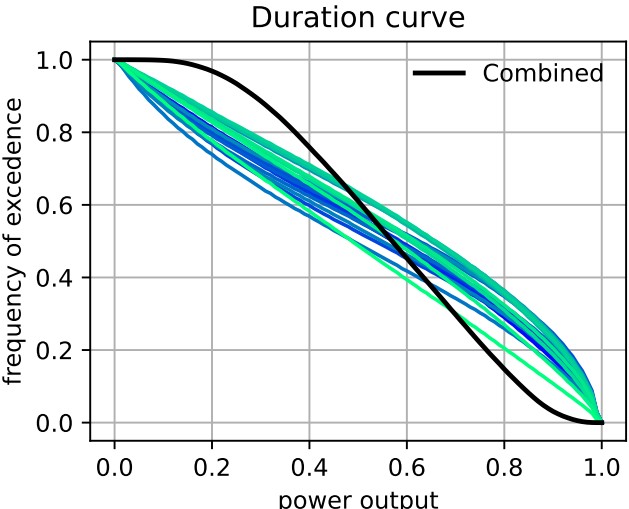

**Figure 6.** Duration curve of wind power output from individual wind farms and combined output

where

$$\sigma_i = \sqrt{\langle (x_i(t) - \langle x_i \rangle)^2 \rangle} \tag{2}$$

is the standard deviation and $\langle \cdots \rangle$ denotes mean value. The correlation coefficient is a normalised measure of covariation that takes values between $-1$ and $1$:





– $c_{ij} = 1$: Full correlation

– $c_{ij} = 0$: No correlation. Statistically unrelated time-series.

– $c_{ij} = -1$: Full anti-correlation, i.e. full correlation in opposite directions

Figure 7 visualises the correlation coefficient matrix $c_{ij}$ for the hourly wind power time series, with values indicated by the colouring. An extract of the full matrix is given in Table 2.

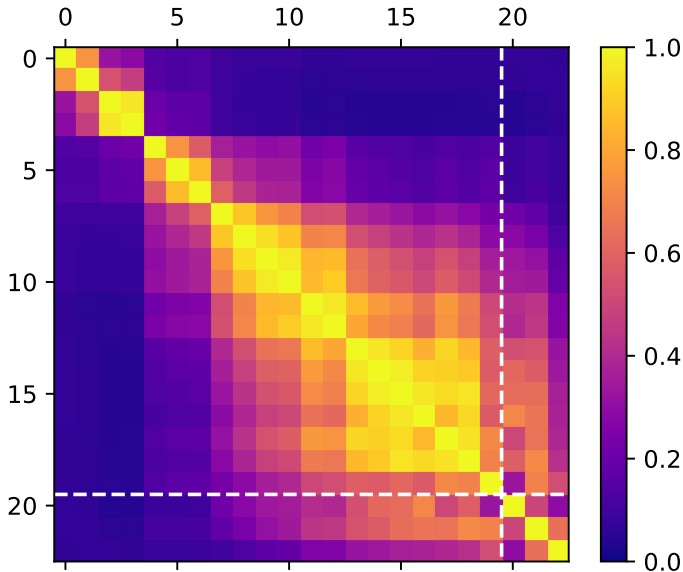

**Figure 7.** Correlation coefficients for hourly power time series for the various wind farm sites. The dashed white line separates the Norwegian wind farm areas to the three other wind farms (20–22).

**Table 2.** Extract from correlation coefficient matrix

| Wind farm sites | Corr. |
|---|---|
| Sørvest F (18) - Nordavind D (3) | 0.05 |
| Sørvest F (18) - Nordvest C (6) | 0.15 |
| Sørvest F (18) - Vestavind F (12) | 0.66 |
| Sørvest F (18) - Doggerbank (20) | 0.58 |
| Sørvest F (18) - Horns rev (21) | 0.72 |
| Sørvest F (18) - Baltic2 (22) | 0.42 |

We see clearly that the correlation reduces with geographical distance, and that there are distinct "wind areas", namely the Nordavind sites (0–3), the Nordvest sites (4–6), the Vestavind sites (7–12), the Sørvest sites (13–18) and Sønnavind (19).





There is low correlation between the Nordavind area and the other sites. This is also the case for the correlation between the Nordvest area and the other sites, albeit not as low as for the Nordavind area. For example, the power output of Sørvest F (18, the site for Sørlige Nordsjø II) has a correlation coefficient of 0.15 with Nordvest C (6) and 0.05 with Nordavind D (3), see also Table 2. The same is true if we consider correlation with European wind farms: For UK-Doggerbank (20) and DK-Horns rev (21), correlation ranges from 0.72 for the Sørvest areas (13–18) to 0.12 and less for Nordvest/Nordavind areas (0–6). Unsurprisingly, correlations with Baltic2 (22) in the Baltic Sea are lower: From 0.52 for Sønnavind (19), 0.42 for Sørvest to less than 0.1 for Nordvest/Nordavind.

Since a large increase in wind power capacity in Danish, German, Dutch, Belgian and British sectors of the southern North Sea is expected, Norwegian wind power that correlates strongly with these areas will have reduced market value compared to wind power that correlates less, since feed-in of large amounts of (wind) generation drives prices down (Hirth, 2013; Sensfuß et al., 2008).

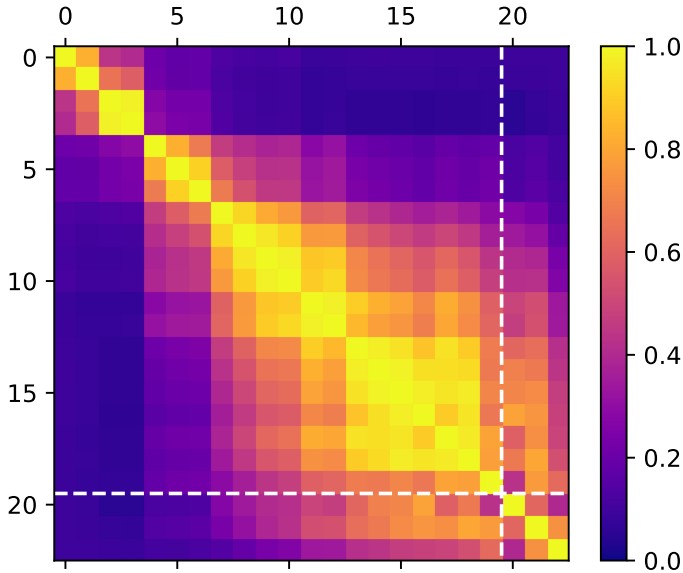

**Figure 8.** Daily time series – correlation coefficients for the various wind farm sites. The dashed white line separates the Norwegian wind farm areas to the three other wind farms (20–22).

These results are based on time series with hourly resolution. It is interesting to explore how the correlation is at different time scales, for example considering daily or weekly averages. Figure 8 shows this for daily and Figure 9 for weekly averages. As can be seen, the correlation coefficient matrices look very similar to Figure 7. This means that the low correlation between e.g. southern North Sea an mid-Norway is there because of weather systems and not (only) short-term random variations in wind. This is consistent with the conclusion in Koestler et al. (2020) that points out that there are different weather systems



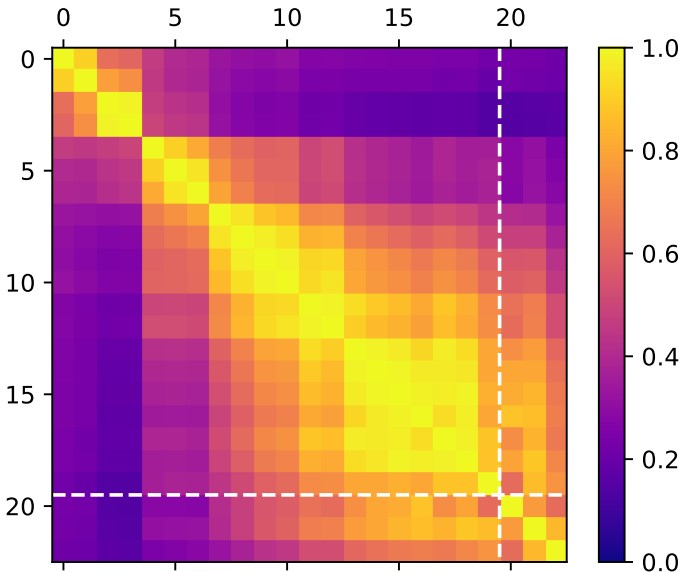

**Figure 9.** Weekly time series – correlation coefficients for the various wind farm sites. The dashed white line separates the Norwegian wind farm areas to the three other wind farms (20–22).

north and south of Stad. Also, we see that there is virtually no correlation between wind in the north and mid-Norway or further south.

## 5 Temporal variability

In this section, we investigate changes in wind power over time, in particular power ramping, low wind period, and seasonal and year-to-year variations of the wind power.

### 5.1 Wind power ramp rates

An aspect of wind power that is very important for power system balancing is the change in power output over time scales up to a few hours. A simple way to characterise this is to compute the *ramps* $R_s$ as the absolute difference in power output $p(t)$ between two instances separated by a time shift $s$:

$$R_s(t) = |p(t_i + s) - p(t_i)|. \tag{3}$$

Figure 10 plots $R_s$ as a filled contour plot with colours indicating the number of occurrences per year for ramps of a certain magnitude during a certain time shift. The non-coloured white area represents values that never occur. Of particular interest are occurrences with high ramps during a short time, since these are the ones that determine the system requirements for compensating actions.





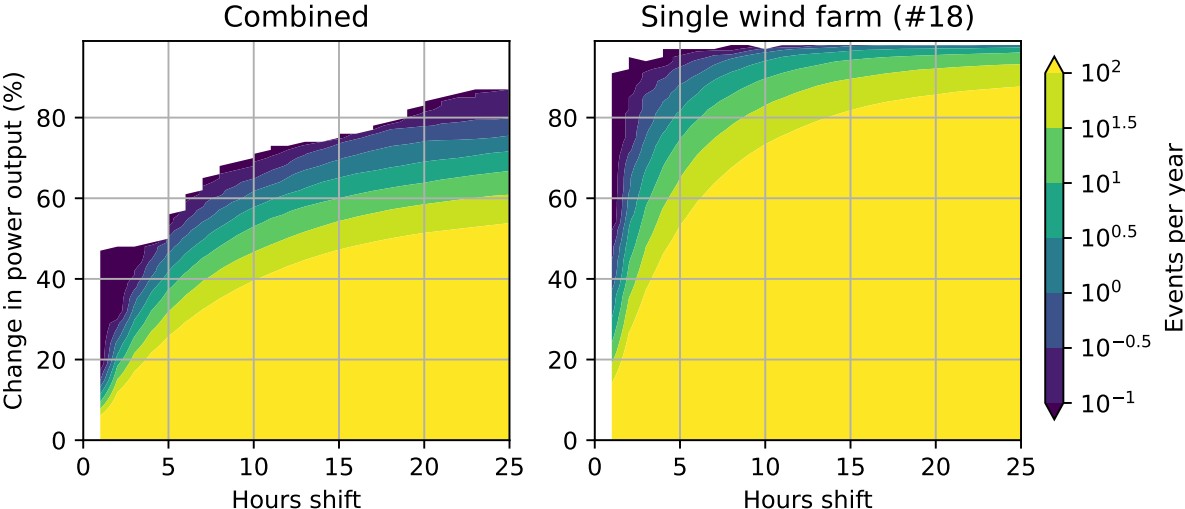

**Figure 10.** Absolute value of change in wind power output after a time shift for combined output (left) and for single wind farm (right). The colours indicate how often the change after a time shift (x-axis) exceeds a certain value (y-axis).

For a single wind farm, the power output can change rapidly. For example, a change of 80% of installed capacity over 5 hours occurs more than $10^{0.5} \approx 3$ times per year. For the combined power output, this type of rapid change never occurs: The shortest time shift to give 80% change is about 18 hours. And the maximum change seen over 5 hours is about 50% of installed capacity.

This smoothing out of the ramp rates is a good thing. It should be kept in mind, however, that this is true only for *relative* values, i.e. compared to installed capacity. The ramps in absolute numbers of course increase with increased capacity, even if they are geographically dispersed. Only if they were to anti-correlate would the opposite happen

Another useful way to illustrate the ramps is in terms of quantiles $q$, defined as the curve that separates the fraction $q$ of situations where $R_s$ exceeds a given value. For example, the 99% quantile curve is defined such that in 99% of the cases the value is below the curve. Figure 11 shows the 99% quantile for $R_s$ a function of the time shift $s$. The plot compares individual wind farms to the combined power output. For example, for a time shift of 5 hours, we see that in 99% of the cases, power output from the individual wind farms change less than about 50–60% of the wind farm capacity, whereas the combined power output changes less than about 25%. In general, the change in the combined power output is almost half that of the change from individual wind farms. In other words, there is a considerable geographical smoothing effect in the ramp rates. The 99% quantile corresponds to 88 time steps per year, i.e. it corresponds roughly to the $10^2$ contour in Figure 10.

High wind power ramp rates are challenging for the power system, especially if they have not been predicted, as they they must be compensated by other generation or load flexibility.

It is interesting to compare wind power ramp rates to ramp rates from power consumption. For this, power consumption data for the period 2015–2021 that are available from the Norwegian TSO have been used (Statnett, 2024). The power demand

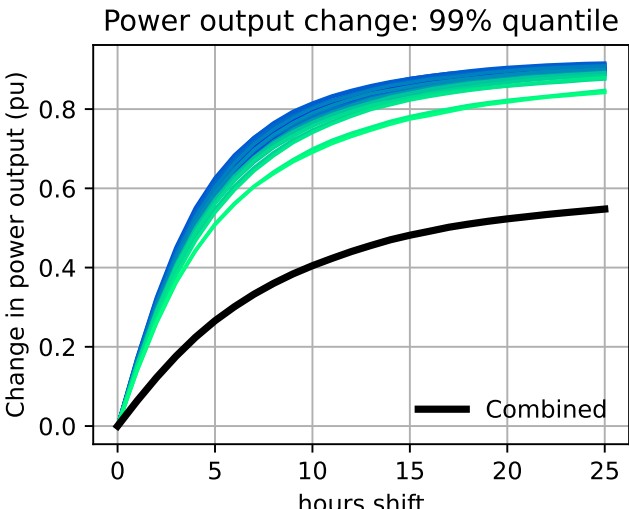

**Figure 11.** Maximum (99% quantile) change in wind power output after a time shift for individual wind farms vs. all combined (black line).

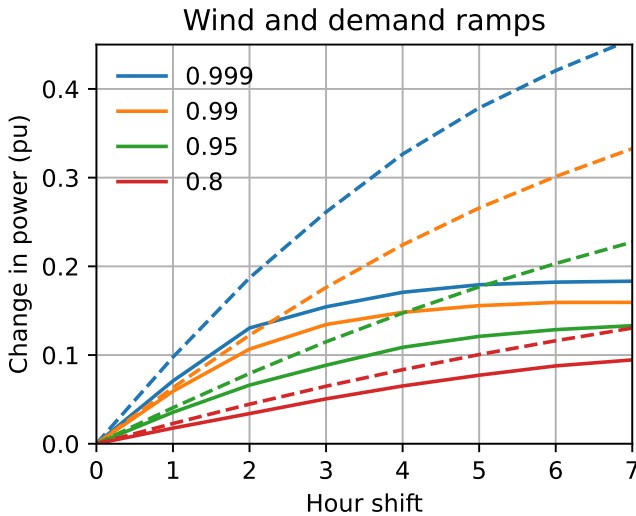

**Figure 12.** Change in combined wind power output (dotted lines) and in power demand (solid lines). Different quantiles.

values have been scaled such that the series has a maximum value of 1. Figure 12 then shows a quantile plot comparing ramp rates for the combined wind power (dotted lines) and power demand (solid line). Due to the daily pattern of power demand, 170 ramping occurs only within a few hours in the morning and in the afternoon. Within these hours, however, the ramping can be high. If we focus on a time window of up to three hours, we see clearly that the ramp rates for the combined wind power





is larger than for the power demand. Yet the differences are on the same scale. The plot includes the 99.9% quantile as an indicator of the worst case ramp rates.

## 5.2 Low wind periods

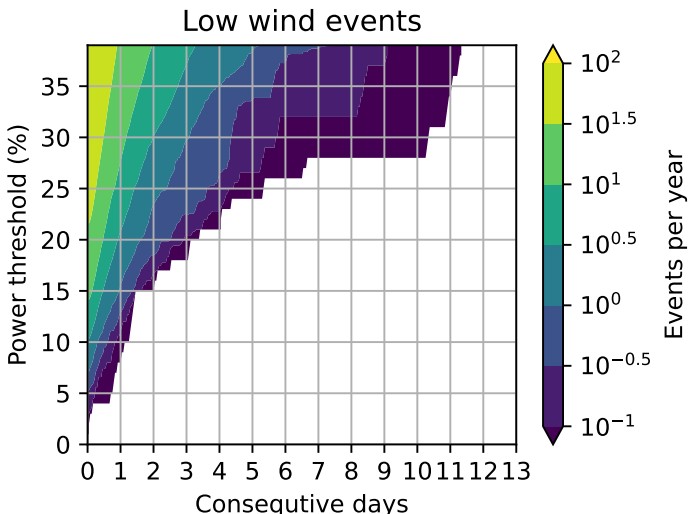

**Figure 13.** Number of events with consecutive hours of wind power below a given threshold

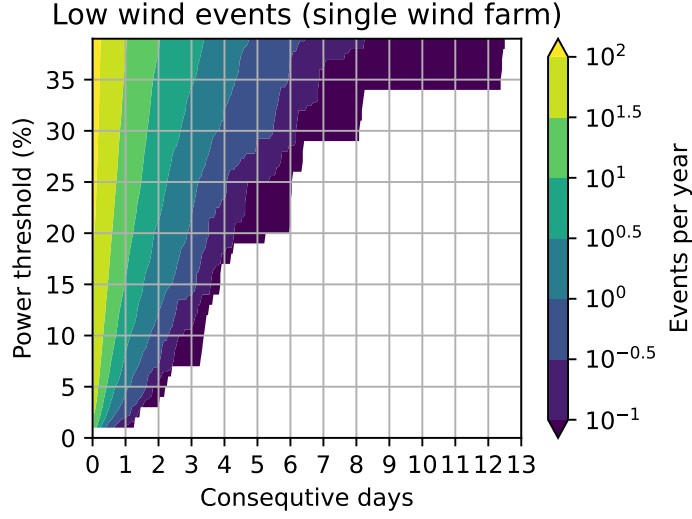

**Figure 14.** Number of events with consecutive hours of wind power below a given threshold



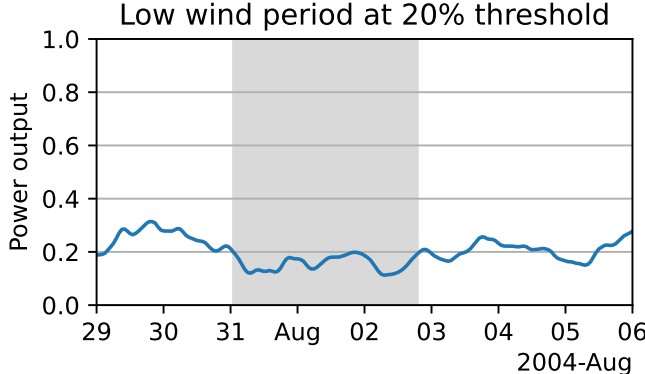

**Figure 15.** Example of extended period with combined wind power output below 20%.

Another critical aspect of wind power is the duration of extended periods with low wind power output (sometimes referred to as "dunkelflaute" events). These are important for understanding the need for alternative power generation capacity, energy storage or demand-side flexibility.

This has been evaluated in the present time series by counting consecutive hours with combined power output below a given threshold value, and then counting the number of such events with prolonged low wind power over the entire time series.

Figure 13 and Figure 14 illustrate this as filled contour plots with colours indicating the number of occurrences per year, for the combined and for a single wind farm respectively. The non-coloured white area represents values that never occur. For example, we see that the maximum time period of combined power output below 20% is about 3 days. The number of events with power output below 20% for more than 1 day is in the range between 1 and $10^{0.5} = 3.16$ per year. Closer inspection of the full data reveals this number to be 75 times for the entire 30 year period, i.e. 2.5 times per year.

The plot clearly shows that a single wind farm has significantly more hours with low output than the combined power. For example, in the combined case, power below 10% never occurs for a duration of more than a little more than a single day, whereas for wind farm site 18, the maximum duration is three and a half days.

An example time-series illustrating a rare low wind period is given in Figure 15. This is an event with power output below 20% for a duration of 66 hours, or 2 days and 18 hours, which from Figure 13 is seen to occur very rarely, about $0.2$ times per
year in these time series. In contrast, Figure 16 shows a period with production sustained above $80\,\%$ of rated power for more than three days, which is a type of event that occurs about $0.4$ times per year (See Figure 17).

These results may be compared with similar analysis by Li et al. (2021) for several European countries.

### 5.3 Seasonal and year-to-year variations

Year-to-year variations in annual wind energy output, as shown in Figure 18, is important for the strategic planning for large
hydropower reservoirs, as larger variations means more year-to-year uncertainty that needs to be considered. The combined annual energy output is seen to vary between $92\,\%$ to $107\,\%$ of the 30 year average value.

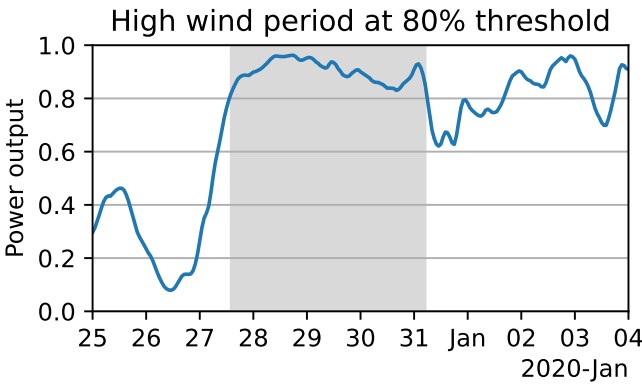

**Figure 16.** Example of extended period with combined wind power output above 80%.

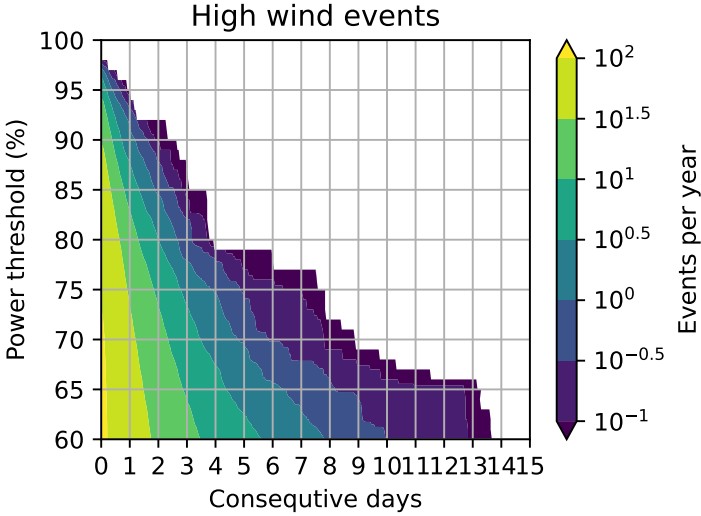

**Figure 17.** Number of events with consecutive hours of combined wind power above a given threshold.





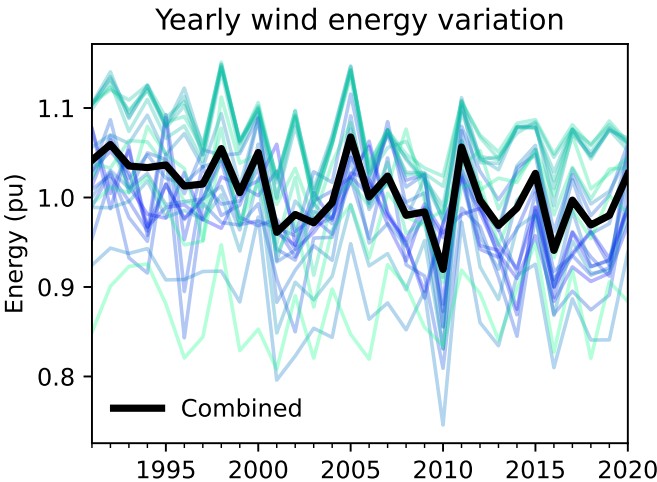

**Figure 18.** Year-to-year variation of the annual energy output of individual wind farms and all combined (thick black line), relative to 30-year average.

Winds in Norway are generally stronger in the winter and therefore wind power production is higher in the winter. This seasonal variation is beneficial since the power demand is also higher in the winter, while water inflow into hydro power reservoirs is low due to winter frost.

Seasonal variation of the combined power output is shown in Figure 19. The data has been grouped according to week number. The solid blue line indicates the average value of the weekly mean power output. The shaded blue area shows plus/minus one standard deviation of the *weakly mean* power for a given weak number for all years. The dashed grey line shows plus/minus one standard deviation of the *hourly* values for a given week number for all years. The dotted line indicates the maximum and minimum hourly value The mean value varies between a maximum of 68% in January and a minimum of 37% in July.

Power consumption data for Norway is available from the transmission system operator, Statnett (Statnett, 2024). The seasonal pattern is illustrated in Figure 20 for the seven years from 2015 to 2021, normalised such that the peak value is 1. The weekly mean value varies between 0.45 in the summer to about 0.77 in the winter. The minimum power demand in the summer is about 60% of the maximum power demand in the winter. The plot includes standard deviations and max/min values as in Figure 19.

Comparing Figure 19 and Figure 20 we see that the seasonal variation of demand and offshore wind power output are very similar when considering their average weekly values. This is a good thing. The variability within a given week of the year, however, is significantly different: Whereas demand varies within a fairly narrow band around the mean value, we see that wind power can have essentially any value between zero and full output.

Seasonal Norwegian power demand and output from 30 GW offshore wind power, together with hydro power inflow (NVE, 2024) is put on top of each other in Figure 21. As already noted, the seasonal pattern of wind power matches very well with the power consumption.

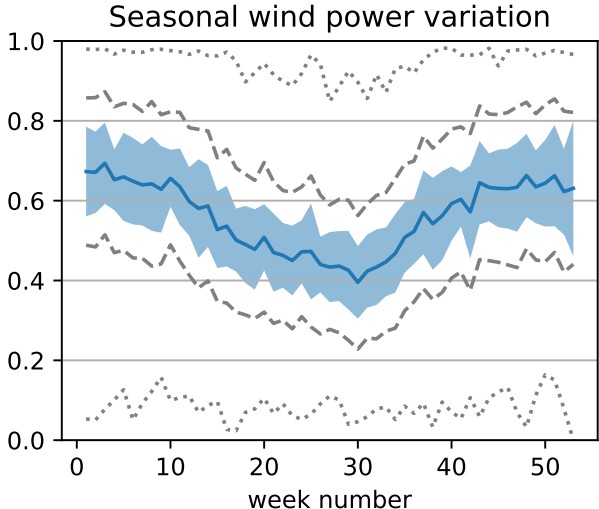

**Figure 19.** Seasonal variation of wind power. Solid blue line shows weakly mean value, shaded area shows plus/minus one standard deviation of weakly mean values, dashed line shows plus/minus one standard deviation of hourly values, and dotted line shows max/min hourly values.

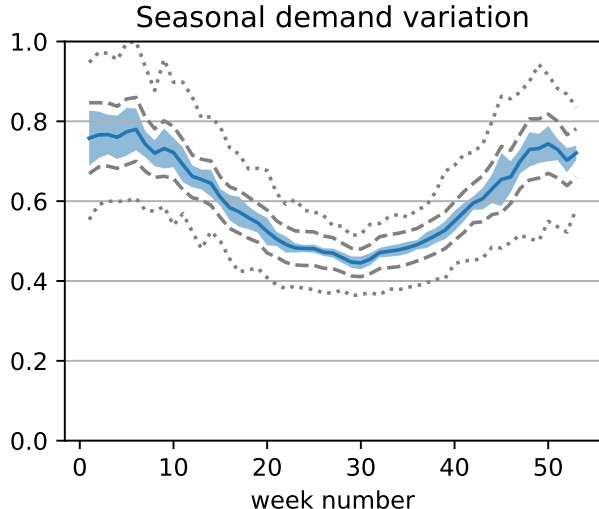

**Figure 20.** Seasonal variation of power demand. Solid blue line shows weakly mean value, shaded area shows plus/minus one standard deviation of weakly mean values, dashed line shows plus/minus one standard deviation of hourly values, and dotted line shows max/min hourly values.

The seasonal variation of offshore wind power is important for the use of Norwegian hydro power and reservoir capacity. And what we see is that the inclusion of large amounts of offshore wind will reduce the need for seasonal balancing by the

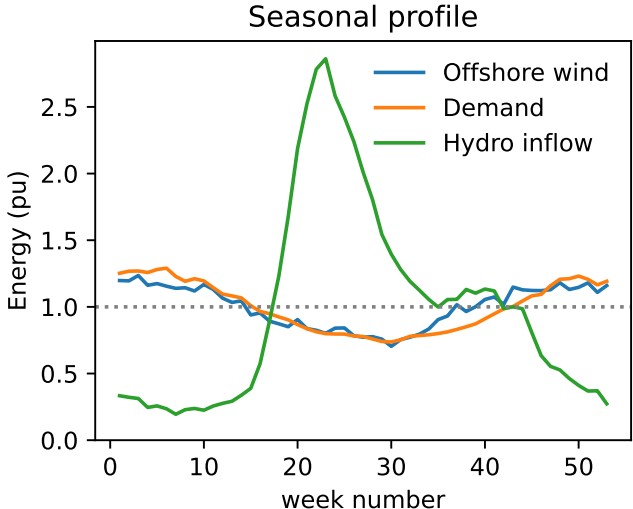

**Figure 21.** Average seasonal profile of 30 GW offshore wind power, hydro power inflow, and power demand (2015-2022)

hydro power system. At the same time, however, the need for balancing on shorter time scales will increase due to wind power
220  variability.

## 6   Conclusion

30 GW of offshore wind in Norway has been found give a combined power output that matches well with the seasonal variation
in the power demand. This will have a big impact on the operation and planning of Norwegian hydro power, and power
exchange with neighbouring countries. But it depends very much on how the power consumption will change in the coming
decades.

We have estimated the power output from publicly available numerical weather model reanalysis data, and have derived
and illustrated various statistical properties that are relevant for investigating the impacts of large-scale offshore wind power
integration in Norway.

Correlation coefficients of wind power output at different potential sites show clearly that correlation are lower with larger
distances, and since large amounts of wind power is expected in the North Sea (mainly in the southern parts), locations from
mid-Norway and northwards have clear advantages.

The 30 GW offshore wind scenario that has been addressed here reflects plans presented by the Norwegian government for
2040. The ambition is to open areas for 2040, meaning that the full 30 GW of installed capacity will come a few years later,
perhaps by 2045 or 2050. At present (2024), this is 21 years into the future – not a long time in energy system planning.
Some final remarks about the approach and futher work are appropriate. The analyses here are based on reanalysis data and
a simple conversion of wind speed to wind power using a power curve representing a wind farm. This misses important wake





effects and site-specific conditions that are required to get reliable yield estimation. At the same time, the future distribution of offshore wind power capacity is largely uncertain, limiting the value of very detailed analyses. Nevertheless, the conversion from a single hourly wind speed data point provided by the Reanalysis data set to wind power output from a geographically

dispersed wind farm is a crucial step that should be improved as much as possible.

Our study here considers offshore wind power only. To give a fuller picture of the impact on the energy system, also other variable renewable energy sources should be included, especially onshore wind power and solar power. But for Norway, 30 GW of offshore wind power is much higher than likely capacity scenarios for onshore wind and solar, justifying their omission in this study.

*Code and data availability.* The code used to download the data and compute the figures presented in this paper is available on the GitHub repository: https://github.com/HaraldGSvendsen/timeseries-analysis (the best place to start is the Jupyter notebook `norway_30GW/norway_30GW_2023areas.ipynb`), (Svendsen, 2024). The wind speed data is available from Renewable.ninja https://www.renewables.ninja/, (Pfenninger and Staffell, 2024).

*Author contributions.* Harald G Svendsen contributed through modelling and analysis, and writing of the paper. John O Tande had the idea

for the paper and contributed in concept development, analysis and writing of the paper.

*Competing interests.* No competing interests are present.

*Acknowledgements.* This activity has been supported financially by the Research Council of Norway via project Ocean Grid Research (project code 328750) and FME NorthWind (project code 321954).



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
