# Peer review of "Analysis of a 30 GW offshore wind power scenario in Norway using time-series computed from numerical weather model data"

_Wind Energy Science, 2024_

## Referee Comment (RC3)

**Analysis of a 30 GW offshore wind power scenario in Norway using time-series computed from numerical weather model data**

by

Harald G Svendsen and John Olav Giæver Tande

The paper analysis the geographical co-variation and variability in offshore wind power production across different time scales. Main results noted in their conclusion are that a combined power output from different sites matches the seasonal variation in the power demand and the correlation between power output from different sites are lower with larger distances.

**General comment**

The article is within the scope of WES, but I find the analysis to be rather superficial and a much more multi-faceted approach is needed to add some novelty to the paper. According to the authors the novelty of the paper lies in the scenario considered (30 GW offshore wind in Norway) but given their rather simple analysis and general conclusions I am not convinced that the manuscript holds the standard needed for publication in WES.

As the results and conclusions rely on output from a rather coarse resolution and more than 10-year old reanalysis system (MERRA 2) which has been shown to perform worse than newer products in other regions, an evaluation of the data should be done to indicate the reliability of the dataset for doing this type of analysis.
As far as I can understand from the paper, the only argument for using MERRA2 is the ease of use. There exist better and more up to date global (ERA5) as well as local (CERRA, NEWA and NORA3) datasets that one should think might be just as good or even better choices. All are freely downloadable.
The paper does not do any relevant evaluation of the MERRA2 data to make sure it is fit for the task. They write that "However, the variability from hour to hour and correlations between different locations, which are our main interests here, are still considered to be well captured by this approach.", but I can not see that this is justified in any way. They do a comparison with actual wind power for one coastal site, but this does not answer to what extent the MERRA2 data mimics real life site-covarability on different time scales. Thus, I will urge the authors to perform a considerably more comprehensive evaluation of the MERRA2 wind speed (and power output if possible), to establish to what extent the data is fit for purpose.

**Specific comments**

- Abstract: The abstract state no results. It is mainly a description of the method and a statement on the relevance of the results for strategic planning.

- Abstract: The abstract begins by stating that "This article investigates implications of integrating 30 GW offshore wind in Norway." This is not really justified. The paper mainly deals with offshore wind analysis and not much with the implications of integrating it into the energy system. The only conclusion they reach on the implications is that the seasonal offshore wind production curve has a similar shape as the seasonal variation in national power demand. I would be surprised if this is not already an established fact for this region.

- Correlation: I find this section rather superficial. What about seasonal differences in co-variability, differences in co-variability during low, medium and high production etc?

- Correlation: Near the end of this result section, the authors speculates on to the marked value of the offshore wind and states that "Since a large increase in wind power capacity in Danish, German, Dutch, Belgian and British sectors of the southern North Sea is expected, Norwegian wind power that correlates strongly with these areas will have reduced market value compared to wind power that correlates less, since feed-in of large amounts of (wind) generation drives prices down" They refer to two general papers about the market value of renewables. Firstly, I am not sure it should be part of the results. Secondly, their speculation might be true, but the authors should justify the claim. A counter argument would be that both now and in the next 30-40 years, onshore wind production will be considerably higher than offshore production. Thus, the price fluctuations related to production fluctuations would possibly be more strongly connected to the large onshore production in countries like Germany than the North Sea offshore production making it problematic to make the general claim that "Norwegian wind power that correlates strongly with these areas will have reduced market value compared to wind power that correlates less"

- Seasonal and year-to-year variations: The authors state that "Comparing Figure 19 and Figure 20 we see that the seasonal variation of demand and offshore wind power output are very similar when considering their average weekly values". I might be mistaken here, but are not the two figures normalized with different values? If so, the fact that the two curves have a similar shape do not mean that "demand and offshore wind power output are very similar" as it does not say anything about to what extent the wind power production can supplement the other production to meet the demand.

- Conclusion: The conclusions are few and only qualitative and the section needs to be improved to reflect the results better. There are no conclusions on several of the result sections such as the low wind analysis and the year-to-year variability results.